# 🔬 SCOPE: SELECTIVE CROSS-MODAL ORCHESTRATION OF VISUAL PERCEPTION EXPERTS

## ABSTRACT

Vision-language models (VLMs) benefit from multiple vision encoders, but naively stacking them yields diminishing returns while multiplying inference costs. We propose SCOPE, a Mixture-of-Encoders (MoEnc) framework that dynamically selects one specialized encoder per image-text pair via instance-level routing, unlike token-level routing in traditional MoE. SCOPE maintains a shared encoder and a pool of routed encoders. A lightweight router uses cross-attention between text prompts and shared visual features to select the optimal encoder from the routed encoders. To train this router, we introduce dual entropy regularization with auxiliary losses to balance dataset-level load distribution with instance-level routing confidence. Remarkably, SCOPE with one shared plus one routed encoder outperforms models using all four extra encoders simultaneously, while reducing compute by 24-49%. This demonstrates that intelligent encoder selection beats brute-force aggregation, challenging the prevailing paradigm in multi-encoder VLMs.

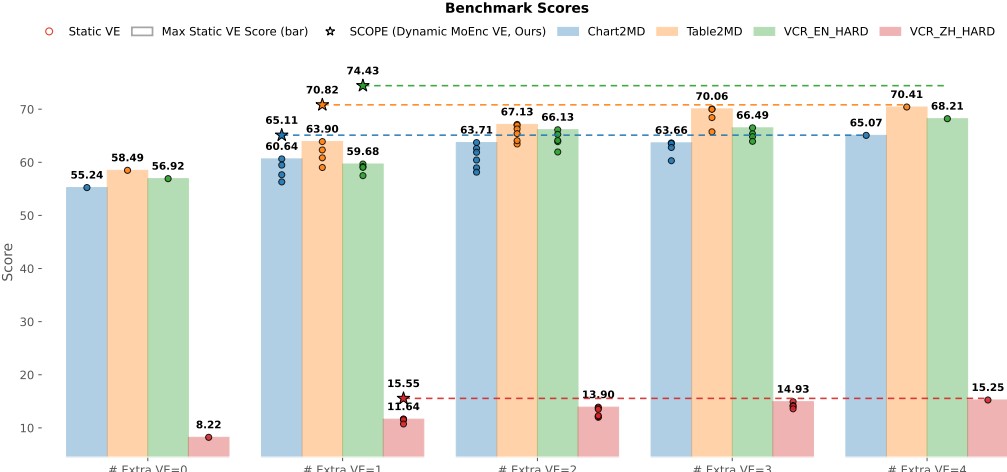

Figure 1: Our model, SCOPE (marked by ⋆), is compared against baseline VLMs configured with zero to four fixed extra vision encoders. For each generation, SCOPE is architecturally equivalent to a single–extra-encoder model but uses a router to dynamically choose that encoder. This dynamic approach allows SCOPE to achieve superior performance across all tasks, notably surpassing the memory-intensive four-encoder model, especially on the VCR_EN_HARD dataset. Please refer to Table 2 for detailed benchmark scores.

## 1 INTRODUCTION

Recent advances in Vision Language Models (VLMs) have demonstrated their remarkable ability to jointly understand and process visual and textual information (Hurst et al., 2024; Comanici et al., 2025; Anthropic, 2025; Bai et al., 2025; Zhu et al., 2025). A promising direction in this field has

been the use of multiple vision encoders to enrich visual representations fed to the large language model (LLM) (Liu et al., 2025; Mao et al., 2025). The rationale behind this approach is that different encoders, pre-trained on diverse datasets and with varied architectures, can capture complementary visual features, leading to a more comprehensive and nuanced understanding of the input image. Several studies have shown the benefits of this multi-encoder approach, reporting improved performance on a range of vision-language tasks (Fan et al., 2024; Kar et al., 2024; Liu et al., 2023b; Shi et al., 2025; Tong et al., 2024; Zong et al., 2024b).

However, the prevailing method of simultaneously deploying multiple vision encoders presents a significant challenge in computational efficiency. The static nature of these multi-encoder setups means that all encoders are activated for every input, regardless of whether their specific expertise is required for the given context. This leads to a suboptimal use of computational resources. In addition, Mao et al. (2025) shows that as the number of encoders increases, the marginal performance gains tend to diminish, while the inference costs, particularly video memory consumption, escalate linearly. Notably, both Mao et al.'s and our observation show that adding a second vision encoder to a single-encoder VLM delivers strong benefits, whereas using more than two encoders offers diminishing returns. This leads to a central question: *When building a VLM for diverse applications, which single additional encoder should we choose?*

To overcome these limitations, we propose a dynamic Mixture-of-Encoders (MoEnc) framework. Our approach SCOPE[1], is motivated by the Mixture-of-Experts (MoE) paradigm (Jiang et al., 2024; Zhou et al., 2022), where a routing mechanism dynamically selects the most relevant *vision encoder* (expert) per input sample. Unlike a standard MoE that often routes at the token level, our MoEnc operates at *instance level*, conditioning the expert choice on both the visual input and the text prompt. In our model, we designate a *shared vision encoder* that is always active and maintain a pool of *routed vision encoders* that remain available. For each inference instance, image / text-prompt pair, a lightweight router dynamically selects exactly one encoder from this pool, whose output representations are combined with the shared encoder before being passed to the LLM. We opt for instance-level routing instead of token-level routing because choosing one expert for the entire image–prompt pair preserves global visual coherence and prevents expert hopping across tokens.

A key innovation in our work is the design of the routing mechanism that employs cross-attention over both *textual prompt embedding* and *visual features* of the shared encoder to select the most suitable routed vision encoder. This design allows the model to adaptively pick the best encoder for each input based on the specific requirements of the input image and prompt, leveraging the strengths of a diverse encoder pool without incurring the computational cost of always using them all. Under this setup, a "1 + 1" configuration (one shared + one routed encoder) can outperform a model that uses all encoders simultaneously, as illustrated in Figure 1.

Remarkably, even if we only utilize image features as the input of the router, the performance remains comparable to the full static all-encoder model. See Table 2 and Section 5 for details.

A central challenge in training our router is a nuanced balancing problem. On the one hand, for effective learning, the router needs to distribute its selections uniformly across the available encoders in the pool over the entire training dataset (load balancing). On the other hand, for a single input, the router should be "confident" in its choice, meaning that the probability of selecting the top-ranked encoder should be substantially higher than the probabilities for the other encoders. To address this, we introduce a novel training strategy incorporating dual entropy regularization and a dual auxiliary loss. This technique successfully reconciles these two competing objectives, leading to a robust and efficient routing mechanism. See Section 2.3.

Our contributions in this paper are threefold:

- We propose a dynamic Mixture-of-Encoders framework that significantly improves computational efficiency while enhancing the performance of VLMs by dynamically selecting from a pool of vision encoders.
- We introduce a novel routing mechanism that utilizes both textual and visual cues to make context-aware encoder selections.
- We present a mechanism that utilizes dual entropy regularization and dual auxiliary loss to effectively address the trade-off between load balancing and routing confidence.

---

[1]The name is inspired by how a microscope selects the appropriate lens for a specimen.

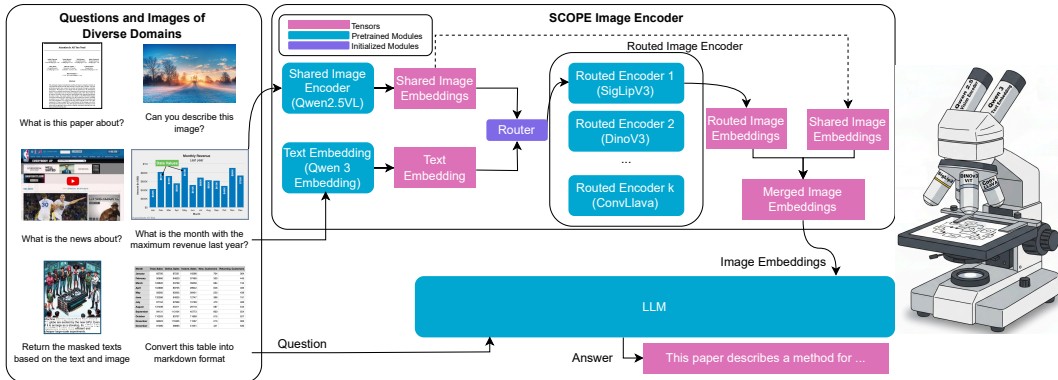

Figure 2: Overview of the dynamic VLM architecture in SCOPE. An input image and user query are processed to generate answers. The architecture features a Shared Image Encoder and a Language Embedding module that generate a Shared Image Embedding and a Language Embedding, respectively. These embeddings are fed into a Router that dynamically selects one Routed Image Encoder from a pool of $K$ expert encoders. The selected auxiliary encoder then generates a Routed Image Embedding. This and Shared Image Embedding is merged into Merged Image Embedding. Finally, this merged embedding is input into a Large Language Model (LLM) to produce the Output Answers. The diagram also highlights that certain modules are Pretrained Modules (cyan), others are Initialized Modules (purple), and the data flowing through the system are Tensors (pink). The dashed lines indicate the selection process, where only one path is chosen. On the right, the microscope serves as a **visual metaphor** for the model's function. It illustrates how SCOPE "zooms in" on a task by routing the input to a selected encoder, akin to a researchers using a microscope to examine a sample with an appropriate objective lens.

## 2 PROPOSED METHOD

Our proposed system introduces a dynamic vision-language processing pipeline that augments a standard VLM with expert selection and feature fusion. The architecture consists of four main stages: initial embedding, router-based expert selection, feature fusion, and final response generation, as shown in Figure 2.

In this section, we present the architecture and training methodology of our dynamic Mixture-of-Encoders framework. Our goal is to create a system that adaptively selects the most suitable vision encoder for a given context, thereby maximizing performance while minimizing computational overhead during inference.

### 2.1 NOTATION

Let $I$ denote the input image, which becomes $I'$ after preprocessing, and let $P$ denote the input text prompt. A frozen text encoder $E_T$ maps $P$ to a representation $T \in \mathbb{R}^{N_T \times D_T}$, where $N_T$ is the number of tokens and $D_T$ the dimension of text embedding.

The SCOPE architecture employs a shared vision encoder $E_S$ that produces an output representation $V_s \in \mathbb{R}^{N_S \times D_S}$ with $N_S$ tokens and feature dimension $D_S$. In addition, SCOPE maintains a pool of $K$ routed vision encoders, $\mathcal{E}_r = \{E_{r_1}, E_{r_2}, \ldots, E_{r_K}\}$, from which exactly one encoder is selected at inference time by a router network $R$. Each routed encoder $E_{r_i}$ produces a representation $V_{r_i} \in \mathbb{R}^{N_r \times D_r}$, which is subsequently passed through a connector network $C_i$ (typically a lightweight linear projection) to obtain an aligned representation $V'_{r_i} \in \mathbb{R}^{N_r \times D_S}$.

### 2.2 SCOPE ARCHITECTURE

Our framework consists of four main stages: (1) Initial Feature Extraction, (2) Dynamic Encoder Routing, (3) Representation Fusion, and (4) Alignment with the Large Language Model (LLM). The overall architecture is depicted in Figure 2.

**Initial Feature Extraction**    Given an input image $I$, we first apply a dynamic resizing preprocessing step that preserves the aspect ratio while limiting the total number of pixels to meet the compute budget. The processed image $I'$ is then fed into the shared vision encoder $E_S$ to obtain the shared visual representation $V_s = E_S(I')$. Simultaneously, the input text prompt $P$ is encoded by the frozen text embedding model $E_T$ to produce the query representation $T = E_T(P)$. The text encoder that we use in our experiments is *Qwen3-Embedding-0.6B*.

**Dynamic Encoder Routing**    At the core of our method is the router module $R$, which dynamically selects a single encoder from the pool of routed encoders $\mathcal{E}_r$. The router leverages both visual and textual cues by employing a cross-attention mechanism in which the query is derived from the query representation $T$ and the keys and values are derived from the shared vision representation $V_s$. The resulting cross-attention output is aggregated into a global representation, which is then passed through a linear layer to produce the routing logits $\mathbf{z} \in \mathbb{R}^K$:

$$\mathbf{z} = \text{Linear}(\text{CrossAttn}(Q = T, K = V_s, V = V_s)). \tag{1}$$

We also consider a variant that omits the text embedding and its corresponding query representation. In this case, the router reduces to a lightweight self-attention mechanism over the shared visual features, followed by a linear projection: $\mathbf{z} = \text{Linear}(\text{SelfAtten}(V_s))$.

**Representation Fusion**    Given the routing logits $\mathbf{z}$, the router performs a top-1 selection by activating only the encoder corresponding to the maximum logit value. Formally, let the selected index be $k = \arg\max_i(z_i)$. At inference time, the preprocessed image is passed through the selected encoder $E_{r_k}$, producing features $V_{r_k}$. Then these are scaled by the corresponding weight $z_k$ and mapped through the connector $C_k$, resulting in the routed representation: $V'_{r_k} = C_k(z_k V_{r_k})$. However, during training, the non-differentiability of $\arg\max$ prevents gradients from flowing directly. To address this challenge, we adopt straight-through estimator (STE) tricks to allow gradients to propagate through the discrete routing decision. Thus, the connector $C_k$ does not receive $V_{r_k}$ directly, instead

$$V'_{r_k} = C_k\left(\sum_{i=1}^{K} z_i V_{r_i} - \mathbf{sg}\left(\sum_{i=1}^{K} z_i V_{r_i} - z_k V_{r_k}\right)\right) \tag{2}$$

where $\mathbf{sg}$ denotes the stop-gradient operator. In this formulation, the backward pass derives gradients from the soft combination $\sum_{i=1}^{K} z_i V_{r_i}$, while the forward pass uses the entries of $V_{r_k}$.

The final visual representation $V_{\text{final}}$ is obtained by concatenating the shared encoder output with the confidence-weighted routed representation:

$$V_{\text{final}} = \text{Concat}\left(V_s; V'_{r_k}\right)$$

**Alignment with LLM**    The fused visual representation $V_{\text{final}}$ is projected into the word embedding space of the LLM, producing a sequence of visual tokens that are prepended to the text prompt embeddings. This visual prefix conditions the LLM on the image content and allows it to perform downstream vision–language understanding tasks.

## 2.3    ROUTER TRAINING WITH DUAL REGULARIZATION AND DUAL AUXILIARY LOSSES

In this subsection, we propose a router training scheme that jointly balances across-batch encoder utilization and per-instance confidence via dual entropy regularizers and complementary auxiliary losses that discourage top-1 collapse while sharpening decisions. We integrate these terms with the language modeling loss using nonnegative weights and show through ablations that this dual-entropy–dual-auxiliary design prevents degeneracy without assuming any scalar relation between batch and instance entropies. In the following, we start by illustrating the challenge of balancing.

A central difficulty during training is to prevent the router from collapsing to a small subset of routed encoders while still making confident, instance-specific decisions. Let $\mathbf{Z} \in \mathbb{R}^{K \times B}$ denote the matrix of routing logits with entries $z_i^{(j)}$, where $i \in \{1, \dots, K\}$ indexes routed encoders and $j \in \{1, \dots, B\}$ indexes samples in a mini-batch.

**Batch balancing via batch entropy and a batch auxiliary loss**  Our first objective is to balance the frequency with which different encoders are activated in a batch. To this end, we introduce *batch entropy regularizer*. For each encoder $i$, we compute probabilities by normalizing along the batch dimension:

$$p_i^{(j)} = \frac{\exp\left(z_i^{(j)}\right)}{\sum_{j'=1}^{B} \exp\left(z_i^{(j')}\right)} = \mathrm{softmax}_j\left(z_i^{(j)}\right), \tag{3}$$

and define the per-encoder batch entropy as $H_{\mathrm{batch},i} = -\sum_{j=1}^{B} p_i^{(j)} \log p_i^{(j)}$. The total batch entropy is $H_{\mathrm{batch}} = \mathbb{E}_{i \in \{1 \cdots K\}} H_{\mathrm{batch},i}$, which we *maximize*. In the loss, we divide it with a normalization factor, which appears as $\mathcal{L}_{\mathrm{be}} = -\frac{H_{\mathrm{batch}}}{\log B}$, encouraging the router to distribute the usage of each encoder more uniformly across the batch.

However, the term $\mathcal{L}_{\mathrm{be}}$ alone is insufficient. Consider an extreme case with $B = 5$, where for every instance $j$ the router produces a nearly uniform distribution with a fixed small bias, e.g. $[0.2 + 4\varepsilon, 0.2 - \varepsilon, 0.2 - \varepsilon, 0.2 - \varepsilon, 0.2 - \varepsilon]$ with $\varepsilon > 0$ tiny. Although $H_{\mathrm{batch}}$ remains high, top-1 routing (we pick $\arg\max_i$ instead of sampling) would still *always* select the same encoder, defeating our balanced design goal. To preclude this degeneracy, we add a *batch auxiliary loss*, inspired by balance auxiliaries in MoE: for each encoder $i$, we form the vector $p_i = [p_i^{(1)}, \ldots, p_i^{(B)}]^\top$ and a one-hot mask $F_i \in \{0,1\}^B$ with its 1 at $\arg\max_j p_i^{(j)}$. We treat $F_i$ with a stop-gradient operator $\mathrm{sg}(\cdot)$ so it is a constant w.r.t. backpropagation. The auxiliary loss is then

$$\mathcal{L}_{\mathrm{ba}} = \sum_{i=1}^{K} \mathrm{sg}(F_i)^\top p_i = \sum_{i=1}^{K} \max_j p_i^{(j)}, \tag{4}$$

which explicitly *minimizes* the largest across-batch probability for each encoder, making it difficult for the router to always select the same instance–encoder pair. We emphasize that $\mathcal{L}_{\mathrm{ba}}$ alone is also inadequate, as it penalizes only the largest entry of each distribution and leaves the rest unconstrained. Thus, we retain $\mathcal{L}_{\mathrm{be}}$ to encourage a balanced non-top-1 load as well.

**Instance confidence via instance entropy and an instance auxiliary loss**  Maximizing $H_{\mathrm{batch}}$ can push the router toward a trivial solution with *uniform* predictions for every instance, indicating that the router has learned little about the instance-specific context. We therefore introduce an *instance entropy regularizer* that acts across the encoder dimension to encourage confident decisions per instance. Define

$$q_i^{(j)} = \frac{\exp\left(z_i^{(j)}\right)}{\sum_{i'=1}^{K} \exp\left(z_{i'}^{(j)}\right)} = \mathrm{softmax}_i\left(z_i^{(j)}\right), \tag{5}$$

and the per-instance entropy $H_{\mathrm{instance},j} = -\sum_{i=1}^{K} q_i^{(j)} \log q_i^{(j)}$. We *minimize* a normalized $H_{\mathrm{instance}} = \mathbb{E}_{j \in \{1 \cdots B\}} H_{\mathrm{instance},j}$ through $\mathcal{L}_{\mathrm{ie}} = \frac{H_{\mathrm{instance}}}{\log K}$, which drives the per-instance distribution over encoders to be sharp.

To further ease optimization, we pair this with an *instance auxiliary loss* that directly rewards the top-1 probability per instance. Let $q^{(j)} = [q_1^{(j)}, \ldots, q_K^{(j)}]^\top$ and $G_j \in \{0,1\}^K$ be a one-hot vector with 1 at $\arg\max_i q_i^{(j)}$, again wrapped by $\mathrm{sg}(\cdot)$. We then define

$$\mathcal{L}_{\mathrm{ia}} = -\sum_{j=1}^{B} \mathrm{sg}(G_j)^\top q^{(j)} = -\sum_{j=1}^{B} \max_i q_i^{(j)}, \tag{6}$$

so minimizing $\mathcal{L}_{\mathrm{ia}}$ maximizes the instance-wise top-1 confidence.

**Combined router objective**  Putting everything together, the router objective combines the language modeling loss with the two entropy regularizers and the two auxiliary terms:

$$\mathcal{L}_{\mathrm{total}} = \mathcal{L}_{lm} + \lambda_{\mathrm{ba}}\mathcal{L}_{\mathrm{ba}} + \lambda_{\mathrm{be}}\mathcal{L}_{\mathrm{be}} + \lambda_{\mathrm{ie}}\mathcal{L}_{\mathrm{ie}} + \lambda_{\mathrm{ia}}\mathcal{L}_{\mathrm{ia}}, \tag{7}$$

with nonnegative coefficients. Note that $\mathcal{L}_{lm}$ is the cross-entropy language model loss. This dual-entropy–dual-auxiliary design reconciles dataset-level load balancing with instance-level confidence. Please note that the two entropies capture fundamentally different axes (across-batch vs. across-encoders) and are therefore not mutually reducible. There is no constant $\gamma$ such that $H_{\mathrm{instance}} \equiv \gamma H_{\mathrm{batch}}$. This can be easily shown by listing 2 examples of $z$ and calculating the corresponding $\gamma$. Besides, we include ablation study results below to show the hyperparameter selection.

## 3 IMPLEMENTATION DETAILS

We adopt the **Qwen-2.5-VL** vision encoder as the shared encoder owing to its native ability to handle inputs with variable spatial resolutions and aspect ratios: a capability that is uncommon among existing vision encoders. The number of parameters in this encoder is 0.67 billion. The text embedding model we choose is **Qwen3-Embedding-0.6B**. The LLM decoder is Qwen-2.5-7B. The routed encoders pool includes:

- **SigLIP 2 (ViT)** **(Tschannen et al., 2025)**: An enhanced version of SigLIP trained with a combination of sigmoid-based language–image alignment loss and a location-aware captioning loss (LocCa (Wan et al., 2024)) on rich multilingual data. This setup makes it well-suited for tasks requiring dense visual features (e.g., document understanding). The size of this vision encoder is 1.14 billion.
- **DINOv3 ViT** **(Siméoni et al., 2025)**: A self-supervised vision transformer trained by self-distillation, where representations of the same image from teacher and student encoders are aligned. Its features are particularly effective for localization and semantic segmentation. The size of this encoder is 0.3 billion.
- **DINOv3 ConvNeXt** **(Siméoni et al., 2025)**: A ConvNeXt variant distilled from the DINOv3 ViT model, offering convolutional inductive biases and efficiency while retaining strong semantic representations. Compared to ViT, ConvNeXt performs better when dealing with high-resolution images. The size of this encoder is 0.3 billion.
- **ConvLLaVA** **(Ge et al., 2024)**: A hierarchical ConvNeXt vision encoder that progressively reduces the token count of high-resolution images into fine-grained representations. By replacing attention layers with convolutions, its computational complexity scales linearly rather than quadratically. The size of this encoder is 0.3 billion.

  Our configuration is designed to ensure both supervisory and architectural diversity. In the routed pool, we include two language-supervised models (SigLIP 2 (ViT), ConvLLaVA) and two self-supervised models (DINOv3 ViT, DINOv3 ConvNeXt). This yields a balanced coverage of ViT and ConvNeXt backbones (two each), offering complementary strengths and reducing bias toward any single paradigm.

The total number of parameters of our model is approximately 11 billion. The activated parameters range from 8 billion to 10 billion depending on the routed encoder selected. From a memory perspective, SCOPE reduces the number of active parameters by 9% to 27% compared to a full multi-encoder setup. To estimate the compute savings, consider an input image of resolution $1024 \times 768$, a text prompt of length 64, and an answer prompt of length 256. Under these conditions, SCOPE yields a 24–49% reduction in compute cost (see Table 1); the detailed calculation procedure is provided in the Appendix B.

Table 1: Compute breakdown and savings (in TFLOPs) for SCOPE with different encoders activated. Lower is better for TFLOPs; higher is better for *Compute Saving*.

|  | + SigLIP2 | + DINOv3 ViT | + DINOv3 ConvNeXt | + ConvLLaVA | + All 4 Encoders |
|---|---|---|---|---|---|
| Shared Enc. | 2.24 | 2.24 | 2.24 | 2.24 | 2.24 |
| Extra Enc. | 2.77 | 1.40 | 1.08 | 1.08 | 6.33 |
| LLM prefill | 8.26 | 7.22 | 7.22 | 4.99 | 9.07 |
| LLM decoding | 1.52 | 1.50 | 1.50 | 1.47 | 1.70 |
| Total | 14.79 | 12.36 | 12.04 | 9.78 | 19.34 |
| Compute Saving | 0.24 | 0.36 | 0.38 | 0.49 | — |

## 4 EXPERIMENTS

### 4.1 MODEL PERFORMANCE AND COMPUTE ANALYSIS

We train our model in two stages with AdamW (learning rate $1e-5$, $\beta_1 = 0.95$, $\beta_2 = 0.999$, weight decay $1e-6$, $\epsilon = 1e-8$) and a cosine scheduler. In the first stage, we update only the MLP connectors and the router using 150K samples from the LLaVA-Pretrain dataset (Liu et al., 2023a). In the second stage, we include the 50K samples from the Arxiv-OCR (nz, 2024) dataset, 30K

samples from Chart2MD, Table2Markdown split in BigDocs (Rodriguez et al., 2024), 50K training samples from VCR (Zhang et al., 2024b), 70K training samples from the DocVQA, InfoVQA, TextVQA and ChartVQA split from DocDownstream (Hu et al., 2024). Among them, Arxiv-OCR is a pure-OCR dataset collected from arXiv.

Chart2Markdown and Table2Markdown is to reconstruct the chart or table's data as a Markdown table. VCR is a task where models must restore partially occluded text in images using pixel-level visual cues and context. DocVQA focus on VQA task on scanned documents that usually require OCR ability. InfoVQA combines reading embedded text with interpreting icons, charts, and diagrammatic elements to answer questions. TextVQA is a VQA task on natural images where answering hinges on detecting and understanding scene text within the image. ChartQA demands extracting plotted values and reasoning over the chart's structure to answer questions. In total, we trained on 200K training samples in the second stage. We trained on the connectors, router and all vision encoders in the second stage. We train SCOPE under the following settings:

- **SCOPE-CA**: the router is a cross-attention mechanism receiving both shared vision encoder representation $V_s$ and text embedding $T$;

- **SCOPE-SA**: the router is a self-attention mechanism receiving only the shared vision encoder representation $V_s$;

- **SCOPE-MLP**: the router is an MLP receiving only the shared vision encoder representation $V_s$;

- **baseline-0**: with no extra vision encoder; the connector is reinitialized for fair comparison;

- **baseline-1**: with a single extra vision encoder; we instantiate this baseline separately with each of the four candidate encoders and report in the table the best performance across these variants;

- **baseline-2**: with two extra vision encoders; we instantiate this baseline for each of the $\binom{4}{2} = 6$ encoder pairs and report the maximum performance across these variants;

- **baseline-3**: with three extra vision encoders; we instantiate this baseline for each of the $\binom{4}{3} = 4$ encoder triplets and report the maximum performance across these variants;

- **baseline-4**: with four extra vision encoders; this is the single configuration with all four encoders active.

The performance of the models mentioned above are listed in the Table 2.

Table 2: Performance comparison of model components across eight benchmarks (higher is better). Bold indicates the best score per column.

| Model | Table2MD | Chart2MD | VCR$_{\text{EN, HARD}}$ | VCR$_{\text{ZH, HARD}}$ | DocVQA | InfoVQA | TextVQA | ChartQA | Avg |
|---|---|---|---|---|---|---|---|---|---|
| SCOPE-CA | **70.8** | **65.1** | 74.4 | **15.6** | **73.1** | **63.4** | **67.9** | 68.3 | **62.3** |
| SCOPE-SA | 68.4 | 63.9 | **75.2** | 14.1 | 70.1 | 59.9 | 65.6 | **68.6** | 60.6 |
| SCOPE-MLP | 67.9 | 62.2 | 70.1 | 13.8 | 69.9 | 60.2 | 64.7 | 67.1 | 59.5 |
| baseline-0 | 58.5 | 55.2 | 56.9 | 8.2 | 65.5 | 52.4 | 61.8 | 64.0 | 53.1 |
| baseline-1 | 63.9 | 60.6 | 59.7 | 11.6 | 67.1 | 56.5 | 64.4 | 65.9 | 56.2 |
| baseline-2 | 67.1 | 63.7 | 66.1 | 13.9 | 68.3 | 57.5 | 65.0 | 66.8 | 58.6 |
| baseline-3 | 70.0 | 63.7 | 66.5 | 14.9 | 69.8 | 59.4 | 65.6 | 67.9 | 59.7 |
| baseline-4 | 70.4 | **65.1** | 68.2 | 15.3 | 69.0 | 60.1 | 65.2 | 67.4 | 60.1 |

## 4.2 HYPER-PARAMETER SELECTION

Our training objective augments the language-modeling loss with two entropy regularization terms and two auxiliary terms. The total loss includes weights $\lambda_{\text{ba}}, \lambda_{\text{be}}, \lambda_{\text{ie}}, \lambda_{\text{ia}}$, where the subscripts denote: $b$ = batch-level, $i$ = instance-level, $a$ = auxiliary, and $e$ = entropy regularization. We explored the hyperparameter settings illustrated in the Table 3. We conclude that both auxiliary loss and entropy regularization help balance the loads. There is a trade-off between the average score and load-balancing when both of them are close to optimum. The best hyperparameter is $\lambda_{\text{ie}} : \lambda_{\text{ia}} : \lambda_{\text{be}} : \lambda_{\text{ba}} = 3 : 3 : 1 : 1$, $\lambda_{\text{ie}}$ could be set ranging from 0.1 to 0.3. Within this range, performance and load-balancing are not sensitive to hyperparameter selection.

Table 3: SCOPE-CA performance across loss-weight configurations. **Avg** is the average task score (higher is better). **Range** is the selection-frequency gap between the most-used and least-used routers (lower is better).

| $\lambda_{be}$ | $\lambda_{ie}$ | $\lambda_{ba}$ | $\lambda_{ia}$ | Avg | Range |
|---|---|---|---|---|---|
| 0.3 | 0.9 | 0.3 | 0.9 | 61.9 | 14.2 |
| 0.2 | 0.2 | 0.2 | 0.2 | 61.1 | 25.8 |
| 0.2 | 0.4 | 0.2 | 0.4 | 62.0 | 16.8 |
| 0.2 | 0.6 | 0.2 | 0.6 | **62.3** | 18.9 |
| 0.1 | 0.3 | 0.1 | 0.3 | 62.1 | 21.5 |
| 0.5 | 0.5 | 0.5 | 0.5 | 59.9 | **9.4** |
| 0.2 | 0.0 | 0.2 | 0.0 | 61.5 | 29.4 |
| 0.0 | 0.6 | 0.0 | 0.6 | 60.2 | 33.8 |
| 0.0 | 0.0 | 0.0 | 0.0 | 55.8 | 98.1 |

## 5 DISCUSSIONS

A practical constraint of our study is that we did not train on the full extent of each dataset due to compute limits. Instead, we used carefully selected subsets. While this choice narrows absolute performance ceilings, it does not undermine our central comparisons: across identical training budgets, the proposed SCOPE framework consistently outperforms static multi-encoder baselines (Table 2). We also included a purely OCR-style English dataset in pretraining, although none of our benchmarks evaluate direct OCR in isolation. Empirically, this addition accelerates optimization and yields better downstream results. We conjecture that exposure to dense text regions improves the model's ability to parse fine-grained, text-rich visual cues that are prerequisites for many VQA tasks.

**VCR results** Performance on VCR$_{ZH, HARD}$ is substantially lower than on the other benchmarks. This gap is expected: our OCR-oriented data are exclusively in English, and the remaining training sets contain no Chinese. However, we retain VCR$_{ZH, HARD}$ because the VCR benchmark probes a distinct ability: attending to small, pixel-level regions that are crucial to answering the question. Notably, SCOPE improves both VCR$_{EN, HARD}$ and VCR$_{ZH, HARD}$, suggesting that dynamic encoder selection helps the model focus on such fine-grained visual evidence even without language-matched supervision.

**When text-conditioned routing helps** We observe clear gains from text-conditioned routing (SCOPE-CA) on DocVQA, InfoVQA, and TextVQA. These tasks exhibit high query diversity: prompts vary not only lexically but also semantically, and the relevant visual regions depend strongly on the question. Conditioning the router on both $V_s$ and $T$ thus appears to be beneficial, probably because different auxiliary encoders have complementary strengths for different visual attributes that the question makes salient.

In contrast, Table2MD and Chart2MD contain prompts that are lexically different but semantically uniform (e.g. 'convert the figure to a Markdown table'). Likewise, VCR uses a fixed query template. In these settings, the value of text features for routing is limited; accordingly, SCOPE-SA (routing from $V_s$ only) can match or even exceed SCOPE-CA. This pattern is consistent with the idea that the router primarily needs text when the question disambiguates which visual features matter.

**Why dynamic selection beats "use everything"** A striking result is that SCOPE-CA consistently outperforms the baseline that fuses all four encoders (baseline-4). At first glance, this is counterintuitive because more visual information should help. In practice, however, aggregating all encoders greatly inflates the number of visual tokens. For Qwen-2.5-VL, a single $1024 \times 768$ image yields roughly 1,036 visual tokens; When multiple encoders are concatenated, the visual prefix can dwarf the textual prompt (often only dozens of tokens). This imbalance lengthens the context and can dilute attention, making it harder for the LLM to identify and reason over the truly relevant evidence. In fact, on Table2MD baseline-3 approaches baseline-4, and on DocVQA and TextVQA baseline-3 even exceeds baseline-4; their average scores (59.7 vs. 60.1) are nearly identical (see Table 2).

Dynamic top-1 selection avoids this overload by admitting only one auxiliary stream, preserving a higher signal-to-noise ratio in LLM's context window.

**Router design: attention vs. MLP** Finally, we note that SCOPE-MLP (a simple MLP router over $V_s$ without text) underperforms SCOPE-SA, despite the common wisdom in MoE that an MLP gate is often sufficient.

**What if batch size per device is 1?** When the batch size is 1, we emulate batch diversity by tiling each image with a LLaVA-Next–style preprocessor: the input is split into $336 \times 336$ patches, which we treat as a pseudo-batch. The router operates per tile, so the dual-entropy/auxiliary objectives are computed across tiles instead of across different images, encouraging balanced expert usage during training; at inference, we apply the same tiling so the router's selections remain evenly distributed. Using this scheme, SCOPE-CA attains an average score of 61.4, with a selection-frequency gap of 22.3, indicating a minor impact on overall performance.

## 6 RELATED WORK

**Vision encoders in VLMs.** Modern VLMs differ mainly in the backbone and pretraining objective of their vision encoders. Contrastive models (CLIP (Radford et al., 2021), SigLIP/SigLIP2 (Zhai et al., 2023; Tschannen et al., 2025)) pair ViT/ResNet backbones with image–text alignment losses. Self-supervised families (DINO/DINOv2/DINOv3 (Caron et al., 2021; Oquab et al., 2024; Siméoni et al., 2025)) scale ViT/ConvNeXt backbones with improved data and training recipes. Task-specialized encoders (e.g., SAM (Kirillov et al., 2023; Ravi et al., 2024) for promptable segmentation; MAE (He et al., 2022) for masked image modeling) provide strong features but with different inductive biases. InternViT (Chen et al., 2024b;a; Zhu et al., 2025) exemplifies a ViT coupled to an LLM via cross-attention, trained with contrastive objectives. Finally, ConvNeXt backbones (ConvLLaVA (Ge et al., 2024)) trade some global context for stronger local spatial modeling and efficient high-resolution processing.

**VLMs with multiple encoders.** Recent systems combine complementary encoders to balance global semantics and fine detail. Two-encoder designs (Janus (Wu et al., 2024), Mini-Gemini (Li et al., 2024), LEO (Lee et al., 2024), Ferret (Zhang et al., 2024a)) pair low- and high-resolution branches, fusing features via interpolation / concatenation, patch-level refinement, or layer-wise cross-attention. Larger mixtures (SPHINX (Lin et al., 2023), Cambrian-1 (Tong et al., 2024)) concatenate or aggregate multigranular features (e.g., with learnable queries in an SVA). Fusion simplicity often suffices: MouSi (Fan et al., 2024) finds MLP projection competitive with Q-Former, and Eagle (Shi et al., 2025) reports that straightforward concatenation of complementary features can match more complex schemes. MoAI (Azadani et al., 2025) and MoVA (Zong et al., 2024a) introduce routing logic to their mixture of module architectures. MoAI precalculates visual, auxiliary, and language features for a learnable router to select; MoVA's routeing technique is not based on learnable routers. Instead, it relies on LLM to classify the task and send the image to the corresponding specialized encoder. The SCOPE router is a learnable module for selecting encoders to process the image features, which is fundamentally different from them.

## 7 LIMITATIONS AND CONCLUSIONS

We introduced SCOPE, a dynamic Mixture-of-Encoders framework that pairs a shared vision encoder with a router-selected auxiliary encoder and trains the router with dual entropy regularization plus auxiliary losses, yielding stronger multimodal reasoning with substantially lower inference cost than static multi-encoder fusion. Across diverse VQA and document understanding benchmarks, SCOPE consistently outperforms baselines, including configurations that activate all encoders, while preserving efficiency by admitting only one routed stream per instance.

**Limitations** Our experiments use subsetted, English-heavy data and we restrict inference to top-1 routing; future work will expand multilingual coverage, explore top-k and token-adaptive routing.

**LLM Usage Declaration** We use LLMs solely for grammar refinement and LaTeX code debugging.

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

## A    PRETRAINED MODEL LIST

In Table 4, we list the pretrained models used in this paper.

Table 4: Pretrained models used and their links.

| Model | Link |
|---|---|
| Qwen-2.5-VL-7B-Instruct | `huggingface.co/Qwen/Qwen2.5-VL-7B-Instruct` |
| Qwen3-Embedding-0.6B | `huggingface.co/Qwen/Qwen3-Embedding-0.6B` |
| SigLIP 2 (ViT) | `huggingface.co/google/siglip2-so400m-patch14-384` |
| DINOv3 ViT (vit-l/16, lvd1689m) | `huggingface.co/facebook/dinov3-vitl16-pretrain-lvd1689m` |
| DINOv3 ConvNeXt (large, lvd1689m) | `huggingface.co/facebook/dinov3-convnext-large-pretrain-lvd1689m` |
| ConvLLaVA-ConvNeXt-1536 | `huggingface.co/ConvLLaVA/ConvLLaVA-ConvNeXt-1536` |

## B    VISION TOKENIZATION AND LLM FLOPS

### B.1    ASSUMPTIONS

- Image size: $1024 \times 768$.
- Text prompt length: 64 tokens.
- LLM (Qwen-2.5 7B): $L = 28$ layers, hidden size $d = 3584$, FFN size $f = 18944$.
- Generation length: $T = 256$ tokens, with KV cache in decoding.
- Prefill length into the LLM: $S_0 = $ #vision tokens to LLM $+ 64$.

### B.2    VISION TOKENS SENT TO THE LLM (PREFILL)

Below, "grid" denotes the spatial token grid before any optional merging; divisions are exact or use ceiling when noted.

**(1) Qwen-2.5-VL native ViT (patch 14, with $2 \times 2$ merge)**

$$\text{grid} = \left\lceil \frac{1024}{14} \right\rceil \times \left\lceil \frac{768}{14} \right\rceil = 74 \times 56, \quad \text{\#ViT tokens} = 74 \cdot 56 = 4144.$$

After $2 \times 2$ spatial merge:

$$\text{\#vision to LLM} = \frac{74}{2} \cdot \frac{56}{2} = 37 \times 28 = 1036, \quad S_0 = 1036 + 64 = 1100.$$

**(2) ConvLLaVA–ConvNeXt-1536 (effective stride $\approx 64$; no extra merge)**

$$\text{grid} = \left\lceil \frac{1024}{64} \right\rceil \times \left\lceil \frac{768}{64} \right\rceil = 16 \times 12 = 192, \quad S_0 = 192 + 64 = 256.$$

**(3) SigLIP2 so400m patch14 (ViT-like, with $2 \times 2$ merge)**

$$\text{grid} = 74 \times 56 = 4144 \quad \text{(patch 14)}, \qquad \text{after } 2 \times 2 \text{ merge} \Rightarrow 1036, \quad S_0 = 1036 + 64 = 1100.$$

**(4) DINOv3–ConvNeXt-L (output stride $32$; no extra merge)**

$$\text{grid} = \frac{1024}{32} \times \frac{768}{32} = 32 \times 24 = 768, \quad S_0 = 768 + 64 = 832.$$

**(5) DINOv3–ViT-L/16 (patch 16, with $2 \times 2$ merge)**

$$\text{grid} = \frac{1024}{16} \times \frac{768}{16} = 64 \times 48 = 3072, \quad \text{after } 2 \times 2 \text{ merge} \Rightarrow 32 \times 24 = 768, \quad S_0 = 768 + 64 = 832.$$

### B.3    LLM FLOPS FORMULAS

We use standard Transformer FLOPs approximations (FP32; one multiply–add $= 2$ FLOPs). For a sequence length $S$ within a layer:

$$\text{Attn proj (Q,K,V,O)}: \ 4Sd^2, \qquad \text{Attn matmuls}: \ 2S^2 d, \qquad \text{FFN}: \ 2Sdf.$$

| Encoder | Vision tokens to LLM | $S_0$ (prefill length) |
|---|---|---|
| Qwen2.5–ViT ($2 \times 2$ merge) | 1036 | 1100 |
| ConvLLaVA–ConvNeXt-1536 (stride $\approx 64$) | 192 | 256 |
| SigLIP2 so400m patch14 ($2 \times 2$ merge) | 1036 | 1100 |
| DINOv3–ConvNeXt-L (stride 32) | 768 | 832 |
| DINOv3–ViT-L/16 ($2 \times 2$ merge) | 768 | 832 |

**Prefill FLOPs (sequence length $S_0$)**

$$\text{FLOPs}_{\text{prefill}} = L \left( 4S_0 d^2 + 2S_0^2 d + 2S_0 df \right).$$

**Decode FLOPs with KV cache (generate $T$ tokens)**   At decoding step $t \in \{1, \ldots, T\}$ the seen length is $S_0 + t - 1$. Per step, per layer:

$$4d^2 + 2(S_0 + t - 1)d + 2df.$$

Summing over $t$ and multiplying by $L$:

$$\text{FLOPs}_{\text{decode}} = L \left( 4Td^2 + 2d \left( TS_0 + \frac{T(T-1)}{2} \right) + 2Tdf \right).$$

**Total LLM FLOPs**

$$\text{FLOPs}_{\text{LLM}} = \text{FLOPs}_{\text{prefill}} + \text{FLOPs}_{\text{decode}}.$$

### B.4   PLUG-IN VALUES (FOR REPLICATION)

For Qwen-2.5 7B alignment used here:

$$L = 28, \quad d = 3584, \quad f = 18944, \quad T = 256, \quad S_0 \in \{1100, 256, 1100, 832, 832\} \text{ per the encoders above.}$$

