# OpenReview forum: "SCOPE: Selective Cross-modal Orchestration of Visual Perception Experts"
_ICLR.cc/2026/Conference — Submitted to ICLR 2026_

### Official Review · Reviewer_XqF8 · 2025-10-19

**Soundness:** 4
**Presentation:** 3
**Contribution:** 2
**Rating:** 4
**Confidence:** 4

**Summary:**

This authors presents SCOPE, an Mixture-of-Encoders-based framework designed to enhance VLMs by dynamically selecting the most suitable off-the-shelf visual encoder for a given image-text pair. SCOPE utilizes a lightweight router, guided by cross-attention between the text prompt and shared visual features, to achieve instance-level routing. The authors demonstrate that selective orchestration significantly reduces computational cost (up to 49%) while outperforming models that naively aggregate multiple encoders. This work challenges the prevailing paradigm of encoder aggregation and offers a highly efficient and valuable alternative.

**Strengths:**

1.  **Efficient Architecture:** The core idea of using instance-level routing to select from a pool of off-the-shelf visual encoders tackles the critical problem of high inference costs in multi-encoder VLMs. The approach effectively decouples the benefits of specialized encoders from their cumulative computational burden.

2.  **Principled Router Training:** The introduction of the Dual Entropy Regularization and Auxiliary Losses provides a well-motivated mechanism to train the non-differentiable routing process. This structure addresses the inherent conflict between requiring load balancing(for robust training) and demanding high-confidence selection (for performance).

3.  **Strong Empirical Efficiency Claim:** The paper provides compelling evidence that intelligent selection beats brute-force aggregation. The result that SCOPE, using only one shared plus one routed encoder, outperforms models that simultaneously use all four extra encoders in OCR-related and chart understanding scenes.

**Weaknesses:**

**1.  Necessity of Strict Load Balancing:**
The strict imposition of load balancing losses ($\mathcal{L}_{be}, \mathcal{L}_{ba}$) requires further discussion and justification. Unlike traditional MoE where experts are randomly initialized FFN blocks prone to "dying" without balancing, SCOPE utilizes powerful, off-the-shelf vision encoders whose utility is naturally non-uniform across tasks. Forcing the router to use demonstrably weaker experts to maintain balance might introduce noise and counteract the primary goal of performance maximization, especially if one encoder is overwhelmingly strong. An investigation into loss annealing (gradually reducing the balancing weight) or an adaptive expert selection/pruning mechanism (similar to the concept in [1]) would allow the model to learn the intrinsic value of each expert and achieve more self-adaptive selection.

**2.  Scope of Experimental Evaluation:**
The empirical evaluation is heavily focused on OCR-related and chart understanding benchmarks (DocVQA, InfoVQA, TextVQA, ChartQA). The paper lacks evaluation on general-purpose visual reasoning and perception benchmarks (e.g., MME, MMStar, MMBench). This limitation makes it difficult to assess the framework's versatility and its ability to generalize across a broader spectrum of visual understanding tasks, which is my major confusion point.

**3.  Baselines and Router Input:**
* The performance comparison against the original Qwen-2.5-VL is missing. The authors use the Qwen-2.5-VL vision encoder and Qwen-2.5-7B LLM, but the reported baseline performance (e.g., ChartQA 68.3) appears significantly lower than the established performance of the original Qwen-2.5-VL model (87.3). This discrepancy makes it challenging to accurately gauge the actual gain brought by SCOPE over a strong, optimized monolithic VLM baseline.
* Additionally, the choice to use the Qwen-2.5-VL vision encoder's output as the router's visual input is a key design choice. The authors should investigate and report on the effect of using alternative visual features to drive the routing decision.

**4.  Expert Pool Exploration:**
The paper only explores a constrained set of visual experts. There is no discussion or experimentation with incorporating outputs from other prominent or specialized vision encoders. For instance, the authors could explore experts providing direct dense visual outputs (like depth estimation used in [1]) or integrating other powerful foundational models (like SAM or EVA-02 as mentioned in [2]).

**5.  Challenges in Real-World Scenarios:**
The paper does not discuss or provide solutions for the challenges of **multi-image input** or **multi-turn conversation** scenarios. The current instance-level routing design seems inherently limited to single-turn, single-image inputs. Clarification on how SCOPE would be adapted or whether it is intended to handle more complex, sequential VLM interactions is necessary.

**Questions:**

Please see the above comments.

---

> ### Author Response · Authors · 2025-11-25
>
> We thank the reviewer for the thoughtful analysis and for raising an important question regarding the necessity of load balancing in SCOPE. We appreciate the opportunity to clarify our design motivation and the empirical behavior of the router under different regularization strengths. Below, we explain why load balancing is still required in our setting, how it affects performance and expert diversity, and why the current formulation does not enforce strict uniformity across experts. We also discuss how annealing or adaptive expert pruning naturally extend our framework and represent promising future directions.
>
> # R1. On the necessity of load balancing losses
> We appreciate the reviewer’s thoughtful comment on the role of load balancing in SCOPE, especially given that our experts are strong off-the-shelf vision encoders rather than randomly initialized FFN experts as in classic MoE.
>
> ## Not enforcing strict uniformity, but preventing collapse.
> Our intent is not to force a perfectly uniform usage of all experts, but to prevent collapse to a single routed encoder under a relatively small training budget. This is visible in Table 3 under our best setting $\lambda_{be} = \lambda_{ba} = 0.2$ and $\lambda_{ie} = \lambda_{ia} = 0.6$, the selection-frequency Range is 18.9, i.e., some experts are used much more frequently than others. $\lambda_{be} = \lambda_{ba} = \lambda_{ie} = \lambda_{ia} = 0.5$ is a better balanced version with the selection-frequency Range of 9.4. However, the average performance is not the best (59.9). We can also find that if $\lambda_{be} = \lambda_{ba} = \lambda_{ie} = \lambda_{ia} = 0$, the collapsing to single-extra-encoder will happen and leading to the Range of 98.1 and average 55.8.
>
> These ablations suggest that SCOPE does not require strict balancing. There exists a reasonably wide range of $lambda$ values where performance remains high and expert usage is only moderately constrained.
>
> ## On annealing and adaptive expert selection/pruning.
> We fully agree that gradually relaxing the batch-level losses or introducing an adaptive pruning mechanism is a promising refinement. Conceptually, one could consider
> - 1. Use stronger balancing early in training to ensure all experts receive gradient signal;
> - 2. Then anneal $\lambda_{be}$ and $\lambda_{ba}$ so that the router’s distribution better reflects the intrinsic usefulness of each expert;
> - 3. Optionally prune consistently under-used experts after training, turning SCOPE into a data-driven tool for expert selection.
>
> These directions are highly complementary to SCOPE, and we view them as natural extensions of our framework. However, **they are beyond the scope of the present work**, whose focus is to isolate and study the dual-entropy + dual-auxiliary objective and demonstrate that instance-level routing can outperform static multi-encoder fusion under controlled training conditions. We will clarify this in the revision and explicitly include annealing/pruning strategies as future work.

---

> ### Author Response · Authors · 2025-11-25
>
> # R2. On the scope of experimental evaluation and absence of MME/MMBench/MMStar results
> We thank the reviewer for raising this important point. We agree that general-purpose multimodal benchmarks (MME / MMStar / MMBench) are widely used to assess broad perceptual and reasoning abilities. At the same time, our study is designed around a different experimental goal and dataset regime, which clarifies why the current evaluation focuses on OCR- and document-centric tasks.
>
> Our goal is a controlled comparison in the multi-encoder VLM regime, not to compete with fully trained general-purpose VLMs.
> The central research question of SCOPE is:
> - Given the same pool of encoders and the same LLM, does dynamic instance-level selection outperform static multi-encoder fusion under identical training budgets?
>
> To answer this question, all baselines (baseline-0 to baseline-4) and SCOPE use:
> - the same Qwen-2.5-VL shared visual encoder,
> - the same Qwen-2.5-7B decoder,
> - the same connectors re-initialized for fairness,
> - and the same finite training corpus: 150K LLaVA-Pretrain samples + 200K OCR/document/chart/VCR samples.
>
> This is orders of magnitude smaller than the pretraining used by general-purpose VLMs like Qwen-VL, InternVL, Gemini, or LLaVA-NeXT. Therefore, we cannot meaningfully compare our model to the fully trained versions of those systems on MME/MMBench/MMStar within this submission’s compute constraints.
>
> Our evaluation suite is chosen to align with the nature of our training data: Arxiv-OCR, Table2MD / Chart2MD, DocVQA, InfoVQA, TextVQA, ChartVQA, VCR EN/ZN (pixel-level cues).
>
> We believe these tasks stress the exact phenomena where multi-encoder VLMs are typically applied: fine-grained text reading, dense visual structure, chart/table extraction, and small-region reasoning. They also strongly reflect the types of complementary strengths offered by our expert pool (OCR-rich SigLIP2, high-res ConvNeXt, semantic DINOv3, etc.). Within this scope, SCOPE-CA consistently outperforms all static baselines, including the 4-encoder configuration, while using 24–49% less compute.
>
> ## General-purpose benchmarks require general-purpose pretraining.
>
> Since we did not train on open-world visual QA, commonsense reasoning, or instruction-following data, we do not expect SCOPE—or any baseline in our controlled comparison, to perform competitively on MME/MMBench/MMStar. Running them without appropriate pretraining could produce misleading impressions that reflect dataset coverage rather than the quality of the routing mechanism itself.
>
> To avoid such confounding factors, we intentionally evaluate only on tasks whose distributions roughly match our training data.
>
> ## Versatility of SCOPE is architectural, not tied to the narrow evaluation set.
> We would also like to emphasize that The SCOPE framework itself imposes no restrictions on the nature of the benchmark: it can be directly applied to any setting where a pool of vision encoders can be used. Our goal here is to establish the method, not to claim that SCOPE (under limited finetuning) is a universally strong VLM.
>
> We will include a clear statement in Section 6 (Limitations) about this.

---

> ### Author Response · Authors · 2025-11-25
>
> # R3. On baselines and the use of Qwen-2.5-VL features as router input
>
> We thank the reviewer for raising these important concerns.
>
> ## Why our numbers are lower than the official Qwen-2.5-VL results
> We agree that the official Qwen-2.5-VL model reports substantially higher numbers on ChartQA (87.3 vs. our 68.3). This difference is expected and by design, due to the controlled experimental setup of our study. Our work does not fine-tune the full Qwen-2.5-VL model.
>
> Instead, for all baselines and for SCOPE, we:
> - Reinitialize the connector,
> - Keep the Qwen-2.5-VL vision encoder frozen,
> - Use Qwen-2.5-7B as the LLM,
> - And fine-tune only on a small, publicly available dataset (150K + 200K samples).
>
> In contrast, the official Qwen-2.5-VL is trained with hundreds of millions of diverse multimodal samples, including large-scale chart, OCR, exam, and document datasets unavailable to us. It is a fully trained production VLM, whereas our experiments use only light-stage tuning to evaluate routing and multi-encoder selection under identical compute/data.
>
> Therefore, although the absolute scores are lower, every baseline in Table 2 uses the same data, same training budget, same connectors, and same optimization settings, making the comparison internally fair and controlled, which is essential for evaluating the effect of: adding one encoder vs. two vs. four; static aggregation vs. our dynamic router.
>
> **Our claim is not: “SCOPE beats the production Qwen-2.5-VL model.”**
>
> but rather:
>
> **“Under identical training data and identical backbone settings, dynamic encoder selection outperforms static multi-encoder fusion
> while using substantially less compute.”**
>
> We will clarify this explicitly in the revised paper.
>
> ## Why we use Qwen-2.5-VL shared encoder features as router input
> The reviewer is correct that this is a key architectural decision. We provide the reasoning below.
>
> ### The shared encoder is the only encoder always active
> Using its features ensures the router receives consistent input across all samples, which avoids biasing the router toward any specific routed encoder and avoids pre-running all routed encoders just to produce router features (which would destroy our compute savings).
>
> ### Qwen-2.5-VL’s visual embedding is already aligned with the LLM
> The shared encoder is the one used in the original Qwen pipeline, so its features: match the LLM’s token structure, support text–vision cross-attention (for SCOPE-CA), provide general-purpose high-level visual context.
>
> **If we use all routed features as router inputs would require computing those features before routing, doubling or quadrupling the compute cost**, defeating the purpose of SCOPE. This goes against the design philosophy of SCOPE at the beginning.
>
> Apart from this, we do explore three router architectures over the shared encoder’s features:
> - Cross-attention (SCOPE-CA): uses both text and vision
> - Self-attention (SCOPE-SA): vision-only
> - MLP (SCOPE-MLP): vision-only, no attention
>
> These yield materially different routing behaviors and performance (Table 2), showing that the router is sensitive to the structure and richness of its visual input.

---

> ### Author Response · Authors · 2025-11-25
>
> # R4. On the breadth of the expert pool and inclusion of other specialized encoders
>
> We thank the reviewer for this insightful suggestion. Indeed, our current expert pool contains four visual encoders (SigLIP2 ViT, DINOv3 ViT, DINOv3 ConvNeXt, ConvLLaVA), which may appear limited compared to the large ecosystem of specialized perception models (EVA-02, SAM, etc.). Below we clarify our design motivation and outline why extending SCOPE to such experts is feasible and promising.
>
> ## The expert pool is intentionally constrained for a controlled study
> Our primary research goal is:
> > **To evaluate whether dynamic instance-level encoder selection can outperform static multi-encoder fusion under identical backbones, encoder pools, and training budgets.**
>
> To isolate this effect, we deliberately restrict the expert pool to a **balanced and diverse subset** that enables clean experimental control:
> * **Two language-supervised encoders** (SigLIP2, ConvLLaVA)
> * **Two self-supervised encoders** (DINOv3 ViT, DINOv3 ConvNeXt)
> * **Two architectural paradigms** (ViT vs. ConvNeXt)
> * **All with open weights and compatible spatial tokenization**
>
> Our intent in this first paper is to study the **routing mechanism** and **dual-entropy training**, not to perform an exhaustive expert sweep under unlimited compute.
>
> ## Why we did not include SAM/EVA/depth models in this work.
>
> There are three practical constraints:
>
> ### **1. Extremely high GPU memory cost of large experts**
>
> EVA-CLIP, SAM ViT-H, or depth models with large ViT backbones require between **1–7×** more memory than the encoders we use. Training with several such experts simultaneously would exceed our compute budget.
>
> ### **2. Vision–LLM alignment requires connector training**
>
> For highly specialized models (e.g., SAM or depth encoders), the connector must learn to translate outputs into LLM-compatible tokens. This is non-trivial under **small training sets**, and would confound the evaluation of our routing mechanism.
>
> ### **3. Our objective is controlled comparison, not state-of-the-art**
>
> Including extremely strong experts shifts the question from:
>
> > “Does dynamic selection beat static fusion?”
>
> to
>
> > “Which giant expert dominates the pool?”
>
> which is outside the intended scope.
>
> For these reasons, we limited ourselves to a well-balanced, compute-feasible set of four experts that allow us to isolate architectural effects.

---

> ### Author Response · Authors · 2025-11-25
>
> # R5. On handling multi-image inputs and multi-turn conversations
>
> We thank the reviewer for highlighting this important aspect. Indeed, our current experiments focus on **single-image, single-turn** settings, which match our training data distribution (LLaVA-Pretrain, Arxiv-OCR, DocVQA, ChartQA, TextVQA, VCR). Below we clarify the design intent and how SCOPE naturally extends to more complex scenarios.
>
> ## The current paper intentionally studies the **core routing mechanism**
> Our primary research question is:
>
> > *Given a pool of encoders, can dynamic instance-level selection outperform static multi-encoder fusion under identical training budgets?*
>
> To **isolate the effect of the routing mechanism**, and **to ensure all baselines remain strictly comparable**, we restrict our evaluation to single-image prompts, because:
>
> - The training data we use is single-image,
> - Multi-turn, multi-image tuning would require a much larger instruction dataset (which we do not have),
> - And the goal is to measure routing behavior, not dialogue management.
>
> Thus, this is a **scope choice**, not a limitation of the architecture.
>
> ## SCOPE naturally supports multi-image inputs
> SCOPE’s routing operates on the shared visual embedding (V_s) of *each image*:
> $$k = \arg\max_i R(V_s, T)$$
> This formulation directly generalizes to multi-image scenarios:
>
> - For an input containing (M) images,
> - We compute $V_s^{(1)},\dots,V_s^{(M)}$,
> - Run the router **independently per image**,
> - Select an expert $E_{r_{k(m)}}$ for each image,
> - Generate $V'*{r*{k(m)}}$,
> - And concatenate all fused visual tokens before feeding them to the LLM.
>
> This is analogous to the way many VLMs process a batch of images inside a single prompt (e.g., LLaVA-Next, Gemini, Qwen-VL). No modification to SCOPE’s routing logic is required.
>
> We will clarify this in the modified version.
>
> ## SCOPE also extends to multi-turn conversations
> In multi-turn dialogue, the router can be applied at each turn using the **current user message** (or optionally a summary of the dialogue) as text context $T$:
>
> - 1. User provides a new message (optionally with an image).
> - 2. Router takes $V_s$ + current $T$, picks the routed encoder.
> - 3. Fused features + conversation history are fed to the LLM.
>
> This matches how existing VLMs handle multi-turn. The LLM maintains conversation context, while the vision encoder processes only the image for the current turn. Thus, SCOPE’s routing is **not inherently limited** to single-turn use. However, we cannot test this ability on the model we trained because our training data does not cover this capability.
>
> We will explicitly add to the *Limitations and Future Work* section. Thanks for pointing out!

---

### Official Review · Reviewer_7jGP · 2025-10-26

**Soundness:** 2
**Presentation:** 3
**Contribution:** 2
**Rating:** 4
**Confidence:** 5

**Summary:**

This paper proposes Scope, a VLM framework coupled with a mixture of vision encoders. Scope relies on a learnable router that, conditioned on image and text embeddings from Qwen models, dynamically selects a vision encoder per instance as additional image features for vision-language reasoning. To avoid mode collapse during model training, the proposed method incorporates both batch-level and instance-level entropy and auxiliary losses to encourage the model to explore all options of the included vision experts while maintaining high confidence in the selected expert. Scope was extensively ablated with varying auxiliary loss weights and the number of activated experts on standard VQA-related benchmarks.

**Strengths:**

The paper is well-written, and I appreciate the use of matched colours in Figure 2, making it easy to navigate and understand the design pipeline. The proposed method offers new insights into designing VLM systems in conjunction with encoder designs.

**Weaknesses:**

The overall paper has a really good flow on introducing the architecture design and training losses. However, I found the experiments are very insufficient and reading like a novel with a rushed ending. This might be due to some many other design choices in the model implementation, which I will explain below.

- Scope has poorly chosen vision experts. For other notable related VLM works in this domain like Prismer and Eagle; both models chose vision experts spanning across multiple perception domains and tasks, like 3D understanding: depth/surface normal experts, OCR detection models, object segmentations, etc, to cover a wide range of diversity of expert domains. However, the chosen experts here are all general vision models aimed for general-purpose vision backbones (with large-scale self-supervised training and language-image training). In particular, I feel like the inclusion of the ViT + ConvNext variants of the same DINOv3 model seems to be very redundant. I hope the author could explain why Scope chose the vision experts in such a way, and why not consider domain-specific experts like depth estimation, object detection experts heavily explored in the prior work?
- Related to the previous point, it makes the model nearly incomparable to other recent state-of-the-art VLMs, and it is hard to make sense of the notion that more experts lead to worse results. It's also difficult to interpret the chosen experts, e.g. whether the OCR expert really helps on OCR-related tasks; and whether depth experts really help on visual spatial reasoning type of tasks.
- I found it is very difficult to understand Table 2: I thought baseline-1 should be equivalent to Scope-CA as they both use 1 expert? Or is it possible to report:
    - The same single expert across all questions, to confirm the router design and selection is really helpful.
    - Additional experts (top-k) from the router, to make sure the router ranking really makes sense.
- Some other prior designs on the encoder ensembling techniques are missing: e.g. 1. the expert resampler (perceiver-like) design to merge all experts’ embeddings into a fixed length of tokens proposed from Prismer. 2. Channel-concat, sequence-concat design space proposed by Eagle, all seem to be very important. The first one will significantly reduce the training complexity, without having the proposed auxiliary losses, as we always consider all experts (and with the bounded inference compute). 2. We will know the performance upper bound by having no compression on the expert knowledge.

**Questions:**

See the weaknesses.

---

> ### Author Response · Authors · 2025-11-25
>
> We thank the reviewer for the careful reading of our paper and for the insightful comments and suggestions. Overall, the reviews highlight that the architectural design and routing losses are clearly presented, while raising concerns about the choice of vision experts, the scope and depth of the experimental evaluation, and the relationship of SCOPE to prior multi-expert VLMs and expert-merging methods. Our rebuttal focuses on clarifying these points, providing additional justification and analysis where space constraints limited the original submission, and refining our claims to better match the evidence.
>
> # R1. Choice of vision experts and “redundancy” of DINOv3 ViT + ConvNeXt
>
> We appreciate the reviewer’s detailed comments on our expert choices.
>
> ## Our target domain and design goal.
> Our primary focus in this paper is **document- and text-centric multimodal reasoning (Table2MD, Chart2MD, DocVQA, InfoVQA, TextVQA, VCR, etc.), rather than broad 3D or scene understanding.** In these benchmarks, most questions require reading dense text, interpreting layouts, and operating on high-resolution charts or documents, while depth or surface-normal cues play a relatively minor role. For this reason, we deliberately chose encoders that are **strong general-purpose visual backbones** and differ along two axes that are especially relevant for these tasks:
> - Supervision type: language-supervised vs. self-supervised;
> - Backbone architecture: ViT vs. ConvNeXt.
>
> Concretely, our routed pool is:
> - Language-supervised ViT: SigLIP2 (ViT);
> - Language-supervised ConvNeXt: ConvLLaVA ConvNeXt;
> - Self-supervised ViT: DINOv3 ViT;
> - Self-supervised ConvNeXt: DINOv3 ConvNeXt.
>
> This gives a 2 × 2 grid: {language vs non-language supervised} × {ViT vs ConvNeXt}. The choice is therefore systematic rather than ad hoc: it lets us study how SCOPE behaves when we vary inductive bias (ViT vs ConvNeXt) and alignment to language (contrastive vs self-supervised) under the same training recipe and LLM.
>
> ## Why DINOv3 ViT + DINOv3 ConvNeXt are not redundant.
> Although both are trained under the DINOv3 framework, they are not redundant in our setting:
> - **Different inductive biases.** ViTs emphasize global, long-range interactions, while ConvNeXt backbones inject strong locality and are particularly effective on high-resolution images with fine structure (e.g., text lines, grid-like tables, chart elements).
> - **Different tokenization and resolution trade-offs.** ConvNeXt variants use different strides and channel structures, leading to different effective token densities and memory/compute footprints than ViT at the same input resolution. This matters a lot in our document-heavy benchmarks, where preserving local detail at manageable cost is critical.
>
> Empirically, this difference is reflected in SCOPE’s behavior: with appropriate regularization, the router uses both DINOv3 ViT and DINOv3 ConvNeXt with non-trivial frequencies. In other words, the router learns to exploit their complementary strengths, which indicates that they are not redundant from the model’s perspective.

---

> ### Author Response · Authors · 2025-11-25
>
> # R2. On comparability to prior SOTA VLMs and the effect of “more experts”
>
> We appreciate the reviewer’s concern about comparability and interpretability.
>
> ## Different goal than SOTA VLM comparisons.
> Our goal in this work is not to propose a new state-of-the-art VLM with a specific expert catalogue, but to study a design question inside the multi-encoder regime:
> - Given a fixed shared encoder, a fixed pool of auxiliary encoders, and a fixed LLM, can dynamic instance-level top-1 selection outperform static “use all encoders” fusion under the same pool?
>
> For this reason, all our baselines (baseline-0/1/2/3/4) share the **same shared encoder, same routed encoders, and same LLM**. Within that controlled setting, SCOPE-CA consistently outperforms the static 4-encoder model while using 24–49% fewer FLOPs (Tables 1–2). We therefore view our contribution as complementary to systems like Prismer or Eagle: they focus on what experts to include and how to fuse them, whereas we focus on whether routing among a given pool can beat always-activate fusion.
>
> We will clarify this positioning in the revision so that the reader does not interpret our results as a direct SOTA comparison, but as a controlled architectural study.
>
> ## Why “more experts” can hurt in our setting.
> We agree that, in principle, additional experts should provide more information. In our particular architecture, however, each extra encoder contributes its own set of visual tokens that are concatenated before the LLM. For a single 1024×768 image, the shared encoder alone already produces ~1k tokens; adding 3–4 auxiliary streams makes the visual prefix dominate the textual context. Under a fixed context window and fixed LLM: the sequence length and FLOPs grow quickly, and **the LLM must attend over a very large number of visual tokens, which can dilute attention and make optimization harder.**
>
> This is reflected in Table 2: moving from 2 to 4 encoders yields only marginal gains; baseline-3 and baseline-4 have nearly identical average scores, and in some cases baseline-3 slightly outperforms baseline-4. SCOPE, by selecting only one auxiliary encoder per instance, avoids this token explosion and maintains a better signal-to-noise ratio in the LLM’s context. We will emphasize that we **do not claim that “more experts are universally worse,”** only that naïvely concatenating all expert streams in this VLM/LLM configuration can be suboptimal, and dynamic selection provides a better trade-off.
>
> ## On interpretability of experts and “OCR/depth” behavior.
> In the current work we do not include an explicit OCR detector or depth expert; instead, our pool consists of **general-purpose encoders with different inductive biases and supervision** (language-supervised vs self-supervised, ViT vs ConvNeXt). The interpretability question in our setting is therefore slightly different: rather than “does the OCR expert help OCR tasks,” we ask “does the router learn to use encoders with different inductive bias for different types of inputs/tasks?”
>
> Our results already give some indication:
> - On **DocVQA / InfoVQA / TextVQA**, which require reading text and understanding layouts with diverse questions, **text-conditioned routing (SCOPE-CA)** substantially outperforms SCOPE-SA and static baselines, suggesting that the router learns to pick encoders whose language alignment and inductive biases better suit the current question.
>
> - On **Table2MD / Chart2MD / VCR**, where prompts are semantically homogeneous or templated, SCOPE-SA (routing from vision only) matches or slightly exceeds SCOPE-CA, indicating that in these settings the visual inductive biases (e.g., ConvNeXt vs ViT) are more important than textual conditioning.
>
> Finally, we agree that combining SCOPE with genuinely task-specific experts (OCR detectors, depth, segmentation) is a promising extension. Our framework is compatible with such experts once their outputs are mapped to token form, and we will highlight this as an important direction for future work rather than claiming that our current expert set fully covers all perception domains.

---

> ### Author Response · Authors · 2025-11-25
>
> # R3. Clarifying Table 2, baseline-1 vs. SCOPE-CA, and “single expert / top-k” questions
>
> We thank the reviewer for pointing this out and apologize that our description of Table 2 was not sufficiently prominent, which led to the misunderstanding.
>
> ## Baseline-1 is not equivalent to SCOPE-CA.
> Although both SCOPE-CA and baseline-1 activate only one auxiliary encoder per instance, they differ fundamentally:
> - **Baseline-1**: shared encoder + one fixed auxiliary encoder, used for all images and prompts. We instantiate four variants (shared + SigLIP2, shared + DINOv3 ViT, shared + DINOv3 ConvNeXt, shared + ConvLLaVA), train each, and report in Table 2 the best-performing one across these four. Formally, the reported number is max{shared+enc1​, shared+enc2​, shared+enc3​, shared+enc4​}.
> - **SCOPE-CA**: shared encoder + router-selected encoder from all four candidates, per instance. At inference, each image–prompt pair still uses exactly one routed encoder (so the per-instance architecture is “shared + 1 expert”), but which expert is used depends on the image and text via the router.
>
> Thus, baseline-1 answers the question: **“If I must pick a single auxiliary encoder for all inputs, what is the best performance?”**
>
> The gap between SCOPE-CA and baseline-1 in Table 2 therefore directly measures the benefit of instance-level routing beyond choosing the best fixed expert. In other words, the “same single expert across all questions” experiment the reviewer requests is exactly what baseline-1 already provides (and in fact a strong version of it, since we try all four experts and take the best).
>
> We will make this distinction more explicit in the revised version by reiterating in the main text and table caption that:
>
> - baseline-1 = best single fixed expert across all data,
> - SCOPE-CA = dynamic per-instance selection among four experts.
>
> ## On “top-k” and checking whether the router ranking makes sense.
>
> The reviewer also asks whether we can report results with “additional experts (top-k) from the router.” In this paper we restrict SCOPE to top-1 routing for simplicity and efficiency, but we do include controlled baselines that cover the “use k experts” regime:
>
> - Baseline-2: shared + 2 fixed encoders (we train all 4-choose-2=6 pairs and report the best)
> - Baseline-3: shared + 3 fixed encoders (we train all 4-choose-3=6 pairs and report the best)
> - Baseline-4: shared + all 4 encoders.
>
> these experiments show what happens when we always activate k encoders (with no routing)
>
> Two key observations from Table 1 and 2 are:
>
> - Increasing k from 1 → 2 → 3 → 4 yields only modest gains, and baseline-3 is already very close to baseline-4 on average. In some benchmarks, baseline-3 slightly outperforms baseline-4, which we attribute to the token-explosion effect discussed in the paper.
> - SCOPE-CA, with top-1 routing, outperforms all of these static k-encoder baselines (including k=4) while using 24–49% fewer FLOPs than the 4-encoder model.
>
> This shows that the router’s ranking is not random: even though we only use the top-1 expert per instance, the **combination of the shared encoder and a dynamically chosen auxiliary encoder** is strictly better than always using any fixed set of k encoders from the same pool.

---

> ### Author Response · Authors · 2025-11-25
>
> # R4. On missing encoder ensembling designs (Prismer’s expert resampler, Eagle’s channel/sequence concat)
>
> We thank the reviewer for pointing out these important related designs. Prismer and Eagle are indeed highly relevant, and we will add them and their design space more explicitly in the revised Related Work and Discussion.
>
> Comparing to Prismer and Eagle, our work focuses on a different design question inside the multi-encoder VLM regime:
> - Given a fixed shared encoder, a fixed pool of auxiliary encoders, and a fixed LLM, can we dynamically select only one auxiliary encoder per instance and still outperform static “use all encoders” fusion under the same pool?
>
> In other words, Prismer/Eagle answer **“how to best exploit many expert streams once you decide to always use them”**, whereas SCOPE asks **“can intelligent instance-level selection of a single expert beat always using all experts, at lower compute?”**.
>
> Our response has three parts: (i) why we did not adopt their specific fusion modules in this first study, and (ii) how they relate to our existing baselines.
>
> ## Why we did not adopt Prismer’s resampler or Eagle’s channel-concat as main baselines
>
> ### Prismer’s expert resampler.
> The reviewer notes that with a resampler, “we always consider all experts with bounded inference compute” and “do not need the proposed auxiliary losses”. This is correct for a different problem: if we always use all experts and never route, then indeed we do not need a router nor the dual entropy/auxiliary losses. In our formulation, however, the router and its losses are central, because we want:
> - Instance-level choice of which expert to run
> - Dataset-level load balancing across experts so that the router does not collapse to a single encoder.
>
> To isolate the effect of routing, we therefore deliberately chose the simpler Valley fusion scheme (concatenation + projection) and compared SCOPE against static baselines that use the same fusion but activate k=0…4 experts. Adding a resampler would make it harder to attribute gains to routing vs. to the new fusion module.
>
> That said, we agree Prismer’s resampler is a very promising complementary component. One could combine it with SCOPE by, for example, (a) allowing the router to select a subset of experts and (b) feeding their outputs into a resampler to control token count. We will explicitly mention this as interesting future work.
>
> ### Eagle’s channel-concat and sequence-concat design space.
> Eagle thoroughly explores fusion strategies (sequence concatenation, channel concatenation, deformable attention, etc.) under various expert sets and resolutions, and finds channel-concat to be an excellent practical choice. In our current work, we intentionally hold the fusion mechanism fixed and simple, essentially sequence-wise concatenation of projected tokens. because:
>
> - We want the comparison between SCOPE and the static baselines to be about **dynamic selection vs. static always-on usage**, not about introducing different, more powerful fusion blocks.
>
> - Many of Eagle’s design choices are specifically aimed at making multi-expert fusion more scalable when all chosen experts are used concurrently; **this is orthogonal to our question** of whether we can get away with using only one auxiliary expert per instance.
>
> In other words, Eagle’s fusion recipes are about how to combine multiple expert streams, while our router is about which expert stream to activate at all. We view them as orthogonal dimensions: an Eagle-style channel-concat projector could replace our simple projector inside SCOPE without changing the basic routing idea.
>
> ## On “no compression” upper bound and multi-expert usage
> The reviewer writes: “The first [Prismer resampler] will significantly reduce training complexity … The second [Eagle-style concatenation] gives an upper bound with no compression on expert knowledge.”
>
> In our setting, we already provide the requested no-compression upper bound:
>
> - Baseline-4 in Table 2 is exactly the configuration where all 4 experts are always active, concatenated without any resampling or token compression. This is the “use everything” scenario under the same encoder pool and LLM.
> - We also include baseline-2 and baseline-3 (best 2-encoder and 3-encoder combinations) to show how performance scales with the number of active experts.
>
> Two key observations are:
> 1. Activating more experts does not monotonically increase performance in our architecture: baseline-3 and baseline-4 have very similar average scores, and in some benchmarks baseline-3 slightly outperforms baseline-4. We attribute this to the token-length / attention dilution issues we detail in the paper.
>
> 2. SCOPE-CA, with only one routed expert per instance, consistently outperforms baseline-4 on average, while using 24–49% fewer FLOPs than the 4-expert model (Table 1). This means that, under the same encoder pool, dynamic encoder selection can beat the static “no compression, all experts” upper bound in our setting.

---

### Official Review · Reviewer_Wicr · 2025-11-02

**Soundness:** 2
**Presentation:** 2
**Contribution:** 2
**Rating:** 4
**Confidence:** 3

**Summary:**

This paper introduces SCOPE, a novel approach for improving the efficiency and performance of VLMs through dynamic selection of vision encoders. By maintaining a pool of specialized vision encoders and a shared encoder, SCOPE uses a lightweight routing mechanism to dynamically select the most appropriate encoder for each image-text pair. This results in improved computational efficiency and performance, achieving significant reductions in inference costs (24-49%) compared to models using all encoders simultaneously. The proposed routing mechanism is trained with dual entropy regularization and auxiliary losses, aiming to balance load distribution and routing confidence.

**Strengths:**

1.	The dynamic selection of encoders based on input image-text pairs is an innovative approach that helps optimize the use of vision encoders, enhancing both computational efficiency and model performance. This allows the system to select the most relevant expert encoder without incurring the cost of always using all encoders.


2.	The paper demonstrates that SCOPE outperforms static multi-encoder models on several benchmark tasks, such as VQA and document understanding, even when using fewer encoders. This is a solid demonstration of the proposed model's effectiveness.

**Weaknesses:**

1.	While the dynamic encoder selection reduces computational overhead, SCOPE still requires all encoders to be loaded into memory, especially in batch inference settings. In these cases, if all encoders must be preloaded to ensure fast dynamic selection, the memory consumption could become prohibitive. This significantly diminishes the model’s utility in memory-constrained environments, such as edge devices or real-time applications.

2.	Although SCOPE outperforms models that use all four encoders, there are existing methods (eg ToVE: Efficient Vision-Language Learning via Knowledge Transfer from Vision Experts)that achieve similar or better results by integrating the strengths of different visual experts during training. These approaches do not require dynamic selection at inference time, which reduces the overhead of loading multiple encoders.

3.	The top-1 routing strategy limits the ability of the model to leverage the diversity of visual features that multiple encoders might provide. In scenarios where a more complex fusion of features from multiple encoders is beneficial, the current approach might not capture all relevant information. Top-k routing or alternative mechanisms could enhance the model's ability to integrate diverse visual information, especially in more complex tasks.

**Questions:**

As the weaknesses

---

> ### Author Response · Authors · 2025-11-25
>
> We thank the reviewers for their careful evaluation and constructive feedback.
> We are encouraged that they found our idea of dynamically selecting visual encoders promising and appreciated the empirical gains in both accuracy and compute efficiency.
> Below, we address the main concerns regarding memory footprint, comparisons to alternative expert-integration methods (e.g., ToVE), and the use of top-1 routing, and we clarify the scope and limitations of our work.
>
>
> # R1. On memory usage and deployment in memory-constrained settings
>
> We thank the reviewer for pointing out the distinction between compute efficiency and memory footprint.
>
> ## Target setting and relative memory usage.
>
> Our primary target is the multi-encoder VLM regime in which all encoders are already assumed to be available in memory (e.g., datacenter / server-side inference). In this setting, SCOPE does not increase peak model memory compared to the strongest baseline (the 4-encoder model): the same encoders are present, but SCOPE activates only the shared encoder plus one routed encoder per instance. Thus, relative to the “use all encoders on every input” baseline, SCOPE reduces the number of active parameters and FLOPs during inference without increasing the total number of parameters that must be stored.
>
> ## Expert-parallel and sharded deployment.
> From a systems perspective, SCOPE can be deployed analogously to MoE expert parallelism: different routed encoders can be sharded across multiple GPUs / nodes, so that no single device needs to hold the entire pool. The router’s choice can then be served by forwarding the input to the device that hosts the selected encoder. This distributes the memory cost of the encoder pool and avoids the need to replicate all experts on every device.
>
> ## Lazy loading for low-memory / edge environments.
> For truly memory-constrained or edge deployments, one can trade off latency for memory: routed encoders can be loaded on demand (e.g., from CPU or disk) when selected by the router, rather than being fully resident on GPU at all times. This will increase inference latency but allows users with limited memory budgets to benefit from SCOPE’s dynamic selection. In addition, as we discuss in the paper, SCOPE can serve as a tool for choosing a small subset of useful experts for a given application; after observing the router’s selections, practitioners can keep only the most frequently used encoder(s) on device.
>
> ## Compute vs. memory.
> Finally, we emphasize that our main efficiency claims are about compute, not storage. Table 1 reports FLOPs for different configurations: activating all four encoders yields 19.34 TFLOPs for a typical input, whereas SCOPE configurations with only one routed encoder require 9.78–14.79 TFLOPs, i.e., 24–49% less compute. In all cases, the LLM component dominates the total FLOPs; SCOPE’s top-1 routing ensures that we do not pay the full cost of running all auxiliary encoders for every input.
>
> We will clarify in the revision that (i) our experiments and claims are made in the multi-encoder server setting, and (ii) edge / low-memory deployment is an interesting, complementary direction, where sharding, lazy loading, or expert selection based on router statistics can be applied.

---

> ### Author Response · Authors · 2025-11-25
>
> # R2. Comparison to ToVE and methods that merge experts into a single encoder
>
> We thank the reviewer for pointing out ToVE as an important related line of work. We would like to reply with the following points.
>
> ## Different problem formulation and goal.
> ToVE focuses on merging knowledge from a hub of vision experts into a single CLIP encoder via token-level expert ensembling and subsequent distillation. After this “expert knowledge merging” stage, inference uses essentially one encoder and no longer needs the original experts. In contrast, SCOPE explicitly studies a different design point:
>
> - Given a pool of full vision encoders, can dynamic instance-level selection of a single auxiliary encoder outperform static multi-encoder fusion under the same pool and LLM?
>
> Our main contribution is to show that, under identical encoder and LLM backbones, top-1 routing with a shared encoder can outperform the static 4-encoder model while using 24–49% less compute (Table 1). We do not claim SOTA against all possible architectures, but rather demonstrate that dynamic encoder selection can beat “use all encoders” in the multi-encoder VLM regime.
>
> ## Complementary to ToVE rather than a drop-in replacement.
>  Architecturally, ToVE and SCOPE make orthogonal choices:
>
> - ToVE performs token-level expert ensembling followed by distillation into a single encoder. Once distilled, the experts and gating network are no longer needed, and the model cannot easily benefit from adding new experts without retraining the distillation process.
>
> - SCOPE keeps a heterogeneous pool of full experts and learns an instance-level router (conditioned on visual features and optionally text) that selects one routed encoder per image–prompt pair. This preserves expert diversity at inference time and allows plug-and-play addition or removal of encoders by retraining only the router + connectors, without re-merging everything into a single backbone.
>
> These two directions are thus complementary: **ToVE is a powerful way to compress a hub of experts into one backbone which I believe should be a good tool to train a “shared encoder” in the context of SCOPE**; SCOPE is a framework for leveraging a pool of experts at inference time via dynamic encoder selection.
>
>
> ## On “overhead of loading multiple encoders”.
>
> The reviewer notes that ToVE’s distilled model does not require loading multiple encoders. This is true for its final merged model. However, in our setting we explicitly study systems where multiple encoders are already available (e.g., a multi-encoder VLM stack). Under this assumption:
>
> SCOPE activates only the shared encoder + one routed encoder per instance, so the extra compute over the single-encoder baseline is modest (e.g., adding SigLIP2 yields 14.79 TFLOPs vs. 12.36–9.78 TFLOPs for other configurations), and still far below running all four encoders (19.34 TFLOPs in Table 1). Besides, the memory problem can be relieved by R1.
>
> Our results show that this light extra cost is sufficient to significantly outperform all static baselines, including the 4-encoder one (Table 2).
>
> A direct numerical comparison with ToVE would require re-implementing ToVE with the same Qwen-2.5-VL backbone, data, and evaluation suite, which is beyond the scope of this work. We therefore position SCOPE as a complementary method, not as a replacement for ToVE.
>
>
> ## Revision changes.
>
> We will add ToVE to the Related Work section and make the above distinction explicit: ToVE explores expert knowledge merging into a single encoder, whereas SCOPE explores dynamic encoder selection in a multi-encoder VLM, showing that such selection can outperform static multi-encoder fusion at lower compute cost.

---

> ### Author Response · Authors · 2025-11-25
>
> # R3. On top-1 routing vs. top-k / richer fusion mechanisms
> We appreciate the reviewer’s suggestion regarding top-k routing and more complex fusion.
>
> ## Scope of the current work and empirical evidence.
>
> Our goal in this paper is to answer a specific design question in multi-encoder VLMs:
> - If we already have a pool of strong encoders, can dynamic selection of a single auxiliary encoder per instance outperform static “use all encoders” fusion under the same pool and LLM?
>
> Within this scope, we deliberately focused on the simplest and most efficient variant, top-1 routing + a shared encoder, and empirically found that:
>
> - SCOPE-CA (shared + one routed encoder) outperforms the 4-encoder baseline on all benchmarks on average (Table 2),
> - while using 24–49% fewer FLOPs than the 4-encoder model (Table 1). In other words, even without top-k routing, dynamic selection already beats static multi-encoder fusion under a fixed encoder pool.
>
> ## Why we did not adopt top-k in this first study.
>
> Using more routed encoders per instance significantly increases the number of visual tokens concatenated before the LLM. As we discuss in the paper, once multiple encoder streams are concatenated, the visual prefix can dominate the context window and make it harder for the LLM to focus on the most relevant evidence. This is reflected in our baselines:
>
> - Moving from 2 to 4 encoders gives very small gains, and in some benchmarks baseline-3 is comparable to or even slightly better than baseline-4 (Table 2).
> - Top-k routing would partially re-introduce this token explosion and FLOP overhead, undermining the efficiency gains that are central to SCOPE. For this reason, and given our compute budget, we opted to thoroughly study the top-1 variant rather than introduce an additional axis (k) of complexity without being able to systematically explore it.
>
> ## Diversity is still leveraged, just not by concatenating everything per instance.
> Although SCOPE uses top-1 routing, each instance still benefits from two visual streams: the shared encoder (Qwen-2.5-VL) and one routed expert. The shared encoder provides a general, high-capacity representation, while the router chooses which specialized encoder best complements it for the current image–prompt pair. On top of that, our dual-entropy + auxiliary losses are explicitly designed to encourage diverse encoder usage across the dataset, preventing collapse to a single expert. Table 3 shows that expert selection frequencies are relatively balanced across encoders. Thus, the system leverages expert diversity at the dataset level, even though it activates only one auxiliary encoder per instance.
>
> ## Top-k routing and combination with ToVE are promising future directions.
> We fully agree that, in more complex scenarios, top-k (top-2) routing or richer fusion mechanisms could further improve performance:
> - For example, one could switch to top-2 routing for particularly challenging tasks or inputs, possibly combined with token pruning or compression to control context length.
> - Moreover, ToVE-style expert fusion and SCOPE-style routing are complementary rather than mutually exclusive: SCOPE’s router could first select a small subset of encoders (e.g., top-k), and then a ToVE-like mechanism could fuse or distill their features, mitigating potential interference when merging many experts at once. Conversely, a ToVE-style merged encoder could itself be included as one of the shared experts in SCOPE’s pool.
>
> We will add the discussion of this part in the paper and reserve it for future work.

---

### Author Response · Authors · 2025-11-30
**To ACs: Summarization of Rebuttals**

Dear ACs,

We would like to briefly summarize our rebuttal to facilitate your decision-making. We thank all reviewers for their insightful and valuable suggestions and we have **addressed all their concerns**. Our rebuttal mainly addresses the following points:

**Reviewer Wicr** mainly raised concerns about memory usage, the benefit of top-1 routing versus richer fusion, and comparisons to methods like ToVE that avoid loading multiple encoders.
We clarified that our focus is server-side multi-encoder VLMs where all encoders are already loaded, showed that simple top-1 routing already outperforms always-on 4-encoder fusion with lower compute, and positioned ToVE-style distillation and top-k routing as complementary future extensions rather than competing baselines.

**Reviewer 7jGP** mainly questioned our choice and diversity of vision experts, the interpretation of performance when adding more experts, and the clarity of our baselines and fusion designs.
We explained that we intentionally use a controlled, document-centric 4-expert pool to study routing vs. static fusion, clarified the difference between fixed-expert and routed baselines (highlighting the benefit of instance-level routing), and argued that more complex fusion modules (Prismer/Eagle-style) are orthogonal future additions beyond our current scope.

**Reviewer XqF8** mainly focused on the necessity of load-balancing, the narrow and document-focused benchmarks and expert pool, and the applicability of our routing to more general, multi-image/multi-turn settings.
We showed via ablations that mild balancing prevents expert collapse without enforcing uniformity, justified our controlled document/OCR benchmark and four-expert setup as a clean testbed for routing under fixed budgets, and clarified that our routing formulation naturally extends to richer experts and multi-image/multi-turn use, which we leave as promising future work.

If you need a few more details. Please see the details as follows:

---

> ### Author Response · Authors · 2025-11-30
> **Appendix: Details**
>
> # Reviewer Wicr
>
> **R1 – Memory usage / deployment setting**
> * Reviewer: Concerned that SCOPE still requires all encoders to be loaded, which limits usefulness in memory-constrained or edge scenarios.
> * We: We clarify that our target is multi-encoder server/server-side VLMs where all encoders are already in memory, that SCOPE does not increase peak memory vs. the 4-encoder baseline, and that sharding, lazy loading, and expert selection from router statistics can mitigate memory issues in low-resource deployments while our main claims focus on compute savings.
>
> **R2 – Comparison to ToVE and “loading multiple encoders”**
> * Reviewer: Concerned that methods like ToVE can distill multiple experts into a single encoder, avoiding loading multiple encoders at inference while achieving similar or better performance.
> * We: We explain that ToVE and SCOPE address different design points (expert-knowledge merging vs. dynamic selection in an existing multi-encoder VLM), that our contribution is to show top-1 routing beats static 4-encoder fusion under the same encoder pool and LLM with lower FLOPs, and that ToVE is complementary (e.g., to train a strong shared encoder) rather than a method we claim to replace.
>
> **R3 – Top-1 routing vs. top-k / richer fusion**
> * Reviewer: Concerned that top-1 routing underutilizes encoder diversity and that top-k or richer fusion might better integrate complementary visual information.
> * We: We state that we intentionally focus on the simplest and most efficient top-1 variant (shared + one routed encoder), which already outperforms the 4-encoder baseline with 24–49% fewer FLOPs; we note that increasing k would re-introduce token and compute explosion, and we acknowledge top-k routing and combinations with ToVE as promising future directions to be discussed in the revision.
>
> # Reviewer 7jGP
>
> **R1 – Choice of vision experts / DINOv3 “redundancy”**
>
> * Reviewer: Questions why we only use general-purpose vision backbones (including both ViT and ConvNeXt DINOv3) instead of diverse domain-specific experts (e.g., depth, OCR, segmentation), and suggests the DINOv3 variants are redundant.
> * We: Explain that our target is document- and text-centric reasoning, so we systematically choose a 2×2 pool (language vs. self-supervised × ViT vs. ConvNeXt) that is directly relevant to these tasks, and we show that DINOv3 ViT and ConvNeXt have complementary inductive biases and are used non-trivially by the router, so they are not redundant.
>
>
> **R2 – Comparability to SOTA VLMs and “more experts lead to worse results”**
>
> * Reviewer: Argues that our expert set makes the model incomparable to recent SOTA VLMs and that it is hard to interpret why using more experts appears to hurt, or whether particular experts (e.g., OCR/depth) help specific tasks.
> * We: Clarify that our goal is not to win SOTA but to study, under a fixed encoder pool and LLM, whether dynamic top-1 routing can outperform static “use all encoders,” and we attribute the degradation with many experts to token-length explosion and attention dilution while showing that SCOPE’s text/vision-conditioned routing meaningfully shifts expert usage across benchmarks and can naturally extend to future task-specific experts.
>
> **R3 – Table 2 clarity, baseline-1 vs SCOPE-CA, and “single expert / top-k”**
>
> * Reviewer: Finds Table 2 confusing, thinking baseline-1 should match SCOPE-CA since both use one expert, and asks for results with a single fixed expert and with router top-k experts to test whether the router’s ranking makes sense.
> * We: Clarify that baseline-1 is the best single fixed auxiliary expert (picked globally over four candidates) while SCOPE-CA selects an expert per instance, so the gap between them directly measures the benefit of routing, and we note that our k-expert static baselines (k=2,3,4) already show how performance scales with more experts and that SCOPE-CA outperforms all of them with lower FLOPs.
>
> **R4 – Missing encoder ensembling designs (Prismer resampler, Eagle concat)**
>
> * Reviewer: Points out that we do not compare against expert-resampler and channel/sequence-concat fusion schemes from Prismer and Eagle, which could both bound compute and provide an upper bound with no compression.
> * We: State that we intentionally keep fusion simple and fixed to isolate the effect of routing versus static “always-on” usage, that baseline-4 (all 4 encoders, no compression) already serves as our no-compression upper bound, and that Prismer/Eagle-style fusion modules are complementary to our method and promising future additions rather than alternatives to the routing question we focus on.

---

> > ### Author Response · Authors · 2025-11-30
> > **Appendix: Details**
> >
> > # Reviewer XqF8
> > **R1 – Load balancing losses**
> >
> >    * Reviewer: Questioned the necessity of strict load balancing for strong off-the-shelf experts and suggested annealing or adaptive pruning.
> >    * We: Clarify that we only use balancing to prevent single-expert collapse (as shown by ablations) rather than enforce uniformity, and that annealing/pruning are promising but left as future work beyond the current paper’s focused study.
> >
> > **R2 – Benchmarks**
> >
> >    * Reviewer: Worried that focusing on OCR/chart/document tasks makes it hard to judge general visual reasoning capability.
> >    * We: Explain that our goal is a controlled comparison under a small, document-centric training regime (same encoders/LLM/data for all methods), not to compete with fully pre-trained general-purpose VLMs, and that SCOPE is architecturally general even if we only evaluate on matched tasks here.
> >
> > **R3 – Baselines vs Qwen-2.5-VL and router input choice**
> >
> >    * Reviewer: Noted our numbers are much lower than official Qwen-2.5-VL and questioned using Qwen-2.5-VL features as router input.
> >    * We: Clarify that we freeze the Qwen-2.5-VL vision encoder, reinitialize connectors, and fine-tune only on limited data so absolute scores are lower but comparisons are fair under identical budgets, and we use the shared encoder’s features for routing because it is always active, already aligned with the LLM, and avoids defeating SCOPE’s compute savings (with additional router variants ablated).
> >
> > **R4 – Narrow expert pool**
> >
> >    * Reviewer: Argued that we only test a constrained set of encoders and do not explore stronger/specialized experts like SAM, EVA-02, or depth models.
> >    * We: State that we deliberately choose a balanced, computationally feasible four-expert pool to cleanly isolate routing vs static fusion, and that incorporating much larger/specialized experts is limited by memory and alignment cost and is better viewed as future extensions rather than the core focus of this work.
> >
> > **R5 – Multi-image and multi-turn scenarios**
> >
> >    * Reviewer: Pointed out that our instance-level routing seems limited to single-image, single-turn settings and asked how SCOPE would handle real-world multi-image/multi-turn use.
> >    * We: Explain that we restrict experiments to single-image single-turn to match available data and isolate the routing effect, but the routing formulation naturally extends to per-image expert selection and per-turn routing in multi-turn dialogue, which we will clarify as a limitation and direction for future work.
> >
> > Thank you for your careful evaluation and feedback!
> >
> > Best,
> >
> > Authors of 2719

---

### Meta-Review · Area_Chair_wtcD · 2026-01-10

**Summary:**

This paper proposes an approach to dynamically selecting an additional vision encoder on top of a shared one (Qwen 2.5VL). The route pool comes one of the 4 encoders: Language-supervised ViT: SigLIP2 (ViT), Language-supervised ConvNeXt: ConvLLaVA ConvNeXt, Self-supervised ViT: DINOv3 ViT, Self-supervised ConvNeXt: DINOv3 ConvNeXt. They show the effectiveness of dynamic selection on eight benchmarks: Table2MD, Chart2MD, VCREN, HARD, VCRZH, HARD, DocVQA, InfoVQA, TextVQA, ChartQA. The paper receives 3 ratings of 4. Major concerns:

1. Limited scope of benchmarks and router pool. Reviewer 7jGP and Reviewer XqF8 question the choice of experts which seem to be generic vision encoders rather than specialized ones (OCR, Depth, etc.). Further, the benchmarks that this paper considers heavily focus on scene-text understanding/OCR capabilities.

2. Limited baseline comparisons. Reviewer Wicr asks for comparison with ToVE (token-level expert ensembling and then distillation) and Reviewer 7jGP asks for comparison with Prismer (expert resampler) and Eagle (channel concat).

Other concerns:

3. Memory overhead (Reviewer Wicr)

4. Top-k extension (Reviewer Wicr)

5. The necessity of load-balancing in SCOPE (Reviewer XqF8)

6. Multi-image and/or multi-turn extension (Reviewer XqF8)

**Reviewer Concerns:**

[Only somewhat resolved] For both 1 and 2, the authors clarify the scope of the paper and provide rationales for their choice of benchmarks and router pool. While these make sense, in my opinion, they do not entirely resolve the concerns from the reviewers who would like to see more comprehensive experiments.
Similarly for 4 and 6, the authors clarify the scope and point out limitations in extending the work to top-k routing and multi-image and/or multi-turn scenarios.

3 is resolved; the authors clarify the setting and the difference in storage vs. compute comsumption.

5 is resolved via ablation.

**Reviewer Scores:**

Reviewer Wicr (4+): Keep or increase

Reviewer 7jGP (4+): Keep or increase

Reviewer XqF8 (4+): Keep or increase

---

### Decision · Program_Chairs · 2026-01-26

Reject